# The Contribution of Motor Changes to Oral Mucositis in Pediatric Cancer Patients: A Cross-Sectional Study

**DOI:** 10.3390/ijerph16183395

**Published:** 2019-09-13

**Authors:** Nyellisonn N. N. Lucena, Lecidamia C. L. Damascena, Isabella L. A. Ribeiro, Luiz M. A. Lima-Filho, Ana Maria G. Valença

**Affiliations:** 1Department of Statistics, Federal University of Paraíba (UFPB), João Pessoa, PB 58051-900, Brazil; lecidamia.damascena@ebserh.gov.br (L.C.L.D.); luiz@de.ufpb.br (L.M.A.L.-F.); anaval@ccs.ufpb.br (A.M.G.V.); 2Department of Public Health, Ribeirão Preto Medical School, University of São Paulo, Ribeirão Preto, SP 14049-900, Brazil; iarrais@ccs.ufpb.br

**Keywords:** child, cancer, motor activity, oral mucositis, logistic regression

## Abstract

This study aimed to investigate the contribution of motor changes to oral mucositis in children and adolescents with cancer undergoing antineoplastic treatment in a referral hospital. This was an observational, cross-sectional study with 70 patients aged 2 to 19 years, diagnosed with any type of cancer and treated in a pediatric hospital cancer ward from April to September 2017. A questionnaire related to the patients’ socioeconomic and clinical conditions was used, followed by the Oral Assessment Guide and selected domains of the activity and participation section of the International Classification of Functioning, Disability, and Health tool. The data were collected by previously calibrated examiners (kappa index > 0.75) and analyzed using descriptive statistics and logistic regression (α = 5%). Children and adolescents aged 7 to 10 years were more likely to develop oral mucositis (OR: 3.62). In addition, individuals who had severe difficulty in maintaining a body position (OR: 14.45) and walking (OR: 25.42), and those diagnosed with hematologic cancers (OR: 6.40) were more likely to develop oral mucositis during antineoplastic treatment. Within the limitations of this study, it is concluded that motor changes may contribute to the occurrence of oral mucositis in pediatric cancer patients.

## 1. Introduction

Cancer remains a public health issue affecting over 14 million people worldwide each year, which corresponds to approximately two-thirds of the deaths in developing countries, outnumbering deaths from HIV/AIDS, tuberculosis, and malaria [1].

Cancer in children and adolescents aged 0 to 19 years accounts for 1% to 3% of all malignant tumors worldwide. It presents clinically distinct from other types of cancer in adults, with short latency periods, good responsiveness to treatment, and good prognosis if diagnosed early [2]. Based on the cancer type, affected organ(s), and risk factors (e.g., age, health status), the treatment protocol may be implemented alone or in combination, which includes surgery, chemotherapy, radiotherapy, and bone marrow transplantation (BMT) or hematopoietic stem cell transplantation (HSCT) [3]. The severity of cancer and its corresponding therapeutic approach may render the patient vulnerable or not to relapse, thereby compromising their quality of life and daily life dynamics [4].

Antineoplastic treatments, especially chemotherapy and radiotherapy, can trigger complications in other tissues and/or organs not originally affected by the disease. Due to the rapid mitotic division of its cells, the oral cavity is commonly affected during the course of antineoplastic treatment, and conditions, such as dry mouth, mucositis, periodontitis, bleeding, gingivitis, candidiasis, as well as fungal, herpetic, and bacterial infections, may occur in susceptible patients [5,6].

Mucositis is the most severe non-hematological disorder caused by antineoplastic treatment [7]. It is characterized by mucosal discontinuity, with inflammation or ulcerative lesions, which may be accompanied by pain, risk of local and systemic infection, bacteremia, and sepsis [8,9]. In the early stage, the mucosa has a whitish appearance but is eventually replaced by an atrophic mucosa, with a swollen, erythematous, and friable aspect. Then, there is evolution to ulcers, with the presence of a removable yellowish fibrinopurulent superficial membrane [10]. Late stages of the condition are characterized by the presence of pseudomembranes formed by dead cells, fibrin, and bacteria [11].

Importantly, discomfort and pain resulting from antineoplastic treatment and development of oral lesions may prolong the patient’s hospital stay, cause nutritional deficit, impaired immune response, and ultimately render them more susceptible to death [12].Other important adverse effects of cancer treatment include changes in physical and psychosocial well-being, cardiovascular and metabolic complications, deficits in physical and cognitive development, as well as the onset of secondary cancers. In pediatric patients, these effects may restrict daily life, compromise motor activities, and cause disability, consequently affecting cardiorespiratory performance, weight, bone density, and hormones (e.g., growth hormone) [13]. In this scenario, physical inactivity during cancer treatment may decrease muscle strength and performance, which added to the clinical picture may result in functional disability and restriction of interpersonal relationships [14,15].

Therefore, understanding the association between the patient’s existing conditions and the proposed treatment protocol is essential for the adoption of more efficient and specific interventions, such as physical exercise, in order to prevent further complications and favor patient adaptation to the hospital setting. This approach may provide benefits in terms of biopsychosocial gains, functional progression, better prognosis, and less difficulties, which should be considered in their habilitation planning.

Given the complications resulting from cancer and antineoplastic treatment, the hypothesis of our study was that motor changes may contribute to the occurrence of oral mucositis in pediatric cancer patients.

Thus, the present cross-sectional study aimed to investigate the association between the presence of motor changes and the occurrence of oral mucositis in children and adolescents with cancer undergoing antineoplastic treatment.

## 2. Materials and Methods

This was an observational, cross-sectional study with pediatric cancer patients undergoing treatment at the Napoleão Laureano Hospital (NLH) and their caregivers. The NLH is a philanthropic organization located in the city of João Pessoa, PB, Brazil, which has been considered a state reference center for the diagnosis and treatment of cancer. The data were collected between April and September 2017. The sample, selected by convenience sampling, consisted of a single group with 70 children and adolescents diagnosed with any type of cancer and admitted to the NLH oncology center (Figure 1).

Patients aged 2 to19 years, diagnosed with any type of malignant cancer, submitted to chemotherapy, radiotherapy, and/or surgery, were considered eligible for this study. The following exclusion criteria were considered: Patients with chronic diseases; those who did not agree to complete the questionnaire; and those whose legal guardians and/or themselves who failed to consent their participation in the study.

This study was approved by the Research Ethics Committee of the Federal University of Paraíba (Universidade Federal da Paraíba—UFPB), under CAAE protocol number 63759516.0.0000.5188. The study followed the Strengthening the Reporting of Observational studies in Epidemiology (STROBE) Statement. Study volunteers (patients and/or legal guardians) consented their participation by signing an informed consent form.

The data were collected using previously validated questionnaires. First, socioeconomic and clinical information were obtained, such as sex, age, marital status, ethnicity, educational level, monthly family income, cancer type, treatment modality, and treatment start date. Second, the Oral Assessment Guide (OAG) was applied to study participants [16]. This instrument considers eight items, which address the voice, the act of swallowing, lips, tongue, saliva, jugal/palate mucosa, and lip mucosa, with scores ranging from 1 to 3, where 1 = normal, 2 = mild/moderate, and 3 = severe, depending on the oral conditions. Third, selected domains of the activity and participation section of the International Classification of Functioning, Disability, and Health (ICF) were applied. Performance and capacity qualifiers were used and quantified according to the following generic scale: 0—no issue; 1—mild issue; 2—moderate issue; 3—severe issue; 4—complete issue. The OAG assessment was performed by calibrated examiners (kappa value > 0.75) in the HNL wards or dental office. The assessment tools required an average of 20 to 30 min to be completed, and the following items were used: A hospital bed and/or reclining chair, a step stool, a lab coat or white coat, gloves, mask and a disposable cap, and flashlights or reflectors. A posteriori sample size calculation was performed. In the logistic regression model, the Wald test statistic is given by the ratio between the maximum likelihood estimate of the parameter and the estimate of its standard error (SE). The test statistics, under the null hypothesis (H0: β_j_ = 0), has standard normal distribution:(1)$$W_{j}=\frac{{\hat{\beta }}_{j}}{SE\left({\hat{\beta }}_{j}\right)}.$$

The *p*-value is defined by:(2)$$p-\mathrm{value}=P(Z>\left|W_{j}\right|),$$
where *Z* follows a standard normal distribution. If the *p*-value was lesser than or equal to α (significance level adopted), the null hypothesis was rejected.

Sample size was determined by Whittemore approximation [17]. According to the prevalence and OR values obtained, for a sample size of 70, we obtained a significance level of 5% (α) and a power of 82.5% (1 − β).

For patients with cognitive deficits, speech deficits, and of a young age, the questions were asked to their caregiver or a proxy informant. To test the assessment tools and methods, a pilot study was previously carried out with five individuals who were later included in the final sample. The data were collected directly from the study participants or from their medical charts, and there was no missing information.

The data were tabulated into Microsoft Office Excel 2010 (Microsoft, Washington, DC, USA) spreadsheets and subsequently transferred to R software (The R Project for Statistical Computing, version 3.3.1 (Free Software Foundation, Auckland, Nova Zelândia). Data analysis was performed using descriptive statistical methods (e.g., absolute and percent frequencies) and inferential statistics via binary logistic regression to checkfor associated variables. Univariate logistic regression analysis was performed for each independent variable (sex; age; ethnicity; educational level; monthly family income; cancer type; treatment modality; altered basic body position; maintaining a body position; self-transferring; lifting and carrying objects; moving objects with the lower limbs; fine use of thehand; use of the hand and arm; walking; moving around; moving around different locations; moving around using equipment; using transportation, such as a car, bus, train, or plane; driving, including riding a bicycle and motorcycle and driving a car; riding animals for transportation; washing oneself, including bathing, drying, and washing hands; caring for body parts, such as brushing and shaving; toileting; dressing; eating; drinking; and caring for own health), considering a statistical significance level of 0.30. After significant variables were identified in the model, the stepwise backward selection method was used to select those capable of explaining the event of interest (α = 0.05). The model fitness was evaluated using the deviance function, and the confounding matrix was used to verify the fitted model through the analysis of hits and misses.

## 3. Results

There was a predominance of males in the sample (54.3%; *n* = 38), with a mean age of 10.9 years (SD: ± 4.90). Most patients were single (97.1%, *n* = 68), with self-reported mixed (52.9%, *n* = 37) and white ethnicity (35.7%, *n* = 25). Approximately 68.6% of the patients (*n* = 48) had elementary schooling, while 8.6% (*n* = 6) of them had no schooling. The monthly family income reported by the caregiver (mother and/or father) was up to two minimum wages (87.1%, *n* = 61) (Table 1).

Most children and adolescents had hematologic tumors (51.4%; *n* = 36), of whom 30.0% (*n* = 21) had acute lymphoid leukemia. Among the solid tumors (48.6%; *n* = 34), osteosarcoma was the most prevalent type, accounting for 14.3% (*n* = 10) of the cases. Chemotherapy was the most frequently used treatment modality, which was reported by 42.9% (*n* = 30) of the individuals, followed by a combination of surgery and chemotherapy (22.9%; *n* = 16). The OAG indicated that 31.4% (*n* = 22) of the patients had oral mucositis, and the main affected sites were the lips (25.7%; *n* = 18), saliva (21.4%; *n* = 15), and gingiva (14.3%; *n* = 10). The ICF assigned performance and capacity qualifiers to the domains (Table 2). In the evaluation of the “mobility” domain, the most frequent impairments regarding “performance” were mild difficulty in walking (14.3%) and moving around (11.4%). Regarding “capacity”, the most frequent impairments were mild difficulty in walking (30.0%) and moving around (28.6%). For the “self-care” domain, the “performance” qualifier indicated a greater occurrence of mild difficulty in caring for body parts (4.3%; *n* = 3) and caring for own health (7.1%; *n* = 5), while the “capacity” qualifier indicated a higher prevalence of moderate difficultyin caring for body parts (32.9%; *n* = 23) and mild and severe difficulty in caring for own health (25.7%; *n* = 18). Other significant impairments were mild difficulty in dressing (24.3%; *n* = 17) and moderate difficulty in washing oneself (20.0%; *n* = 14).

Each independent variable was initially submitted to simple logistic regression analysis in order to identify those with greater association with the onset of oral mucositis. Subsequently, the variables showing statistical significance were included in the final model (*p*-value < 0.3) (Table 3).

The final model selected the variables of cancer type, maintaining a body position, and walking (*p*-value < 0.05). The variable age was also included due to its importance in explaining the presence or absence of oral mucositis and for showing a *p*-value close to 0.05 (Table 4).

As shown in Table 4, individuals aged 7 to 10 years were 3.6-fold (maximum 5-fold) more likely to develop oral mucositis when compared to those of other age ranges. In addition, individuals who had severe difficulty in maintaining a body position and walking were 14.4-fold (maximum 16.3-fold) and 25.4-fold (maximum 28.2-fold) more likely to present oral mucositis, respectively, as compared to those with other difficulties. Lastly, patients diagnosed with hematologic cancer were 6.4-fold more likely to develop oral mucositis during antineoplastic treatment as compared to those with solid tumors.

The deviance function was used to verify the validity of the model since the deviance statistic for the adjusted logistic regression model (63.11448) is lower than the Chi-square reference value (84.82065). The ROC curve was generated to ensure the quality of the model, which can measure the probability of a dichotomous response based on a cutoff point. In our study, probability estimates below and above the cutoff point corresponded to patients without/with oral mucositis, respectively. The cutoff point was 0.439, with sensitivity (true positives) of 54.5%, specificity (false positives) of 89.6%, and an area under the curve of 0.804 or 80.4%, indicating a good fit. Next, a confounding matrix (contingency table) was constructed to identify the hits and misses of the model when determining the occurrence of mucositis. A cutoff point of 0.439 was considered, which resulted in a total of 78.57% of correctly classified cases.

Despite the number of explanatory variables, the model can be used as a reference to predict the factors associated with the onset of oral mucositis due to motor changes, since, after adjustment, more than 10 events remained per variable, maintaining satisfactory model quality.

## 4. Discussion

The results of the present study demonstrated that males were most frequently affected by oral mucositis, which is consistent with international [15,18,19] and national reports [6,20]. Although younger individuals were the majority in the sample, age varied according to the type of cancer [13,21].

In our study, most individuals self-reported their ethnicity as mixed, agreeing with a Brazilian study conducted with children and adolescents in Teresina, Piauí [22], which may be explained by the considerable miscegenation of the Brazilian population [23]. Most participating children and adolescents were attending elementary school during treatment, as expected for this age group [22]. Hematologic tumors were the most frequent cancer type in the study sample, with a predominance of acute lymphoid leukemia (ALL) [5,15,24,25,26], which is considered the most common neoplasia in this age group [27]. Leukemias have been reported to affect children more frequently and may account for 25% to 35% of all childhood tumors [2]. Regarding anticancer treatment, chemotherapy was the most common treatment modality regardless of whether it was used alone or in combination with other treatment protocols. These data are consistent with previous studies carried out in the South [28] and North regions of Brazil [25]. This therapeutic modality associated with chemotherapy has been responsible for the high rate of cure and associated with increased patient survival [28].

Antineoplastic treatment may cause several side effects, with oral mucositis being one of the most frequent complications affecting the oral cavity [5,6,19]. Our study showed that 31.4% of the children and adolescents presented oral mucositis, which is consistent with previous prevalence studies with the same age group conducted in Brazil (37%) [6] and Hong Kong (24%) [26]. Other studies have indicated even higher prevalence rates of oral mucositis in the pediatric cancer population [19,20,21].

The clinical condition of individuals with cancer may be worsened due to the occurrence of oral mucositis, which is considerably painfuland tends to reach higher grading as thecondition aggravates [29]. This can result in food or liquid intolerance, limiting oral nutrition and causing dehydration, malnutrition, electrolyte disturbance, and the need for parenteral diet [30]. In the present study, there were no cases in which children with oral mucositis needed a parenteral diet as all of them were on an oral diet.

Most patients had functional complications (e.g., in playing activities) that could influence and modify their routines, which may be a consequence of changes in muscle strength, aerobic resistance, motor coordination, bone density, and consequently, functional mobility [31,32]. Some studies mentioned limitations imposed by treatment of hematologic [33] and solid tumors [34], among which are impaired mobility and disturbance in daily lifeactivities. Our study further indicated some deficits in the self-care ability of children and adolescents with cancer in performing activities, such as bathing, brushing teeth, dressing, and undressing, which affects their functionality within the space where they live.

Age is an important factor in the development of self-care skills, whose learning starts in the first years of life and expands over time [35]. A greater number of young individuals were included in ourstudy, which may have favored the identification of deficits as they are still developing their self-care skills. The onset of oral mucositis was significantly associated with the age group of 7 to 10 years, that is, children within this age rangewere more likely to develop oral mucositis. This is directly related to the fact that, in addition to treatment and tumor characteristics, there are also particularities, such as early age and poor oral cavity hygiene, contributing to oral lesions [10]. These characteristics may aggravate the oral health of cancer patients, since younger individuals are more prone to develop oral lesions, which can be present in up to 90% of the cases under the age of 12 [6,21]. Importantly, all patients included in our study (and their parents/caregivers) had access to educational actions (oral hygiene instruction and information about oral mucositis) as well asto preventive measures (topical application of fluoride).

The patient’s condition, such as the presence of dental caries and poor oral hygiene, can result in severe and prolonged oral complications during cancer treatment [36]. Therefore, oral health is an essential factor for controlling these issues, with an emphasis on oral fitness, maintenance of oral hygiene, and control of opportunistic infections.

Another association observed in our study was between the diagnosis of hematologic tumors in children and adolescents and the occurrence of oral mucositis. This relationship was previously observed in national [20] and international studies [19]. Hematologic tumors can be considered a risk factor for the development of oral mucositis and immunosuppressant based on the frequency of chemotherapy [10].

With regard to motor changes, severe difficulty in maintaining a body position was significantly associated with the onset of oral mucositis. Children and adolescents with cancer felt limited in performing some functional activities due to excessive care from their families and multiprofessional team [37]. Other factors that restrict patients to a hospital bed, such as adverse effects of treatment and hospital medical devices used during hospitalization, should be further explored [13,31]. A possible hypothesis is that, when facing such a difficult scenario, parents and/or caregivers have a desire to maintain the child’s comfort and rest, especially when they are debilitated due to treatment, which may limit the development and independent execution of motor skills, such as eating, walking in the hospital, and performing personal hygiene tasks. Such limitations indirectly affect other conditions of the patient, such as the practice of oral hygiene, increasing the likelihood of oral mucositis.

While poor oral conditions may contribute to aggravating oral mucositis, oral hygiene practice can prevent this complication and even accelerate the healing process when it has been already initiated [11,12,13,14,15,16,17,18,19,20,21,22,23,24,25,26,27,28,29,30,31,32,33,34,35,36,37,38]. Based on our findings, individuals affected by cancer with severe difficulty in walking had a higher probability of developing oral mucositis. Several modifications in the patients’ bodies may contribute to greater musculoskeletal inactivity and a sedentary lifestyle [13,32]. In addition, children and adolescents with cancer showed altered motor coordination, muscle flexibility, and muscle speed and strength, which are essential for routine activities. One study related suchalterations with the patient’s difficulty in walking and possible dependence on caregivers [15].

Different types of cancer and antineoplastic treatments may impact one’s ability to perform simple activities, such as walking or playing games [39]. Motor changes induced by cancer may gradually weaken children and adolescents during treatment. Consequently, they tend to be restricted to a bed and require help from third parties, which may negatively impact their body immunity, contribute to the development of infection/sepsis, and favor the occurrence of oral mucositis. In addition to treatment and oral health conditions, immunological markers (inflammatory and immunosuppressive mediators) can also influence the occurrence of oral mucositis [38].

This study has important limitations to consider: (i) The cross-sectional design of the study does not allow monitoring changes during antineoplastic treatment; therefore, no causality can be inferred from the outcomes [40]; (ii) other variables that could contribute to the onset of oral mucositis, such as the patient’s oral health, dosage of chemotherapeutic drugs, hematological parameters, and renal function, were not considered. Further research should include these variables to provide a comprehensive view of associated oral complications, particularly of those related to motor changes in children and adolescents undergoing antineoplastic treatment; (iii) this study was conducted in a single hospital, not as a multicenter initiative, which reflects a smaller sample sizedue to the low incidence of pediatric cancer [41], restricting the possibilities to extrapolate the results [42].

Despite the limitations, this study provides mounting evidence on the contribution of motor changes to the occurrence oral mucositis in children and adolescents underdoing antineoplastic treatment.

## 5. Conclusions

Within the limitations of this study, it is concluded that the age range of 7 to 10 years, diagnosis of hematologic cancer, and severe difficulty in maintaining a body position and walking increase one’s likelihood of developing oral mucositis.

## Figures and Tables

**Figure 1 ijerph-16-03395-f001:**
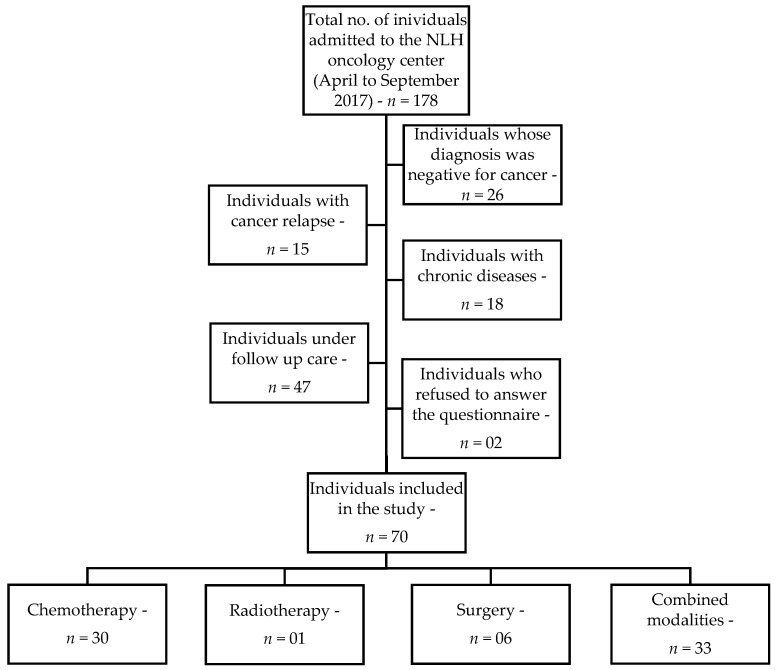
Flowchart of study sample selection.

**Table 1 ijerph-16-03395-t001:** General characteristics of the study sample. João Pessoa, PB, Brazil, 2017.

Variable		(*n*)	(%)	Presence of Oral Mucositis	
				(*n*)	(%)
Sex	Male	38	54.3	13	18.6
	Female	32	45.7	9	12.8
Age	07 to 10 years	19	27.1	8	11.4
	Other age ranges	51	72.9	14	20.0
Marital Status	Single	68	97.1	21	30.0
	Married	2	2.9	1	1.4
Race/Ethnicity	Mixed	37	52.9	9	12.8
	White	25	35.7	8	11.4
	Indigenous	4	5.7	1	1.4
	Black	4	5.7	4	5.7
Educational Level	Did not attend formal education	6	8.6	1	1.4
	Elementary school	6	8.6	1	1.4
	Middle school	48	68.5	19	27.1
	High school	10	14.3	1	1.4
Monthly Family Income	Up to 2 minimum wages	61	87.1	20	28.6
	2 to 6 minimum wages	9	12.9	2	2.8
Cancer Type	Hematologic	36	51.4	15	21.4
	Solid Tumor	34	48.6	7	10.0
Treatment Modality	Chemotherapy	30	42.8	15	21.4
	Radiotherapy	1	1.4	1	1.4
	Surgery	6	8.6	1	1.4
	Chemotherapy+ Radiotherapy	8	11.4	1	1.4
	Chemotherapy + Surgery	16	22.9	2	2.8
	Chemotherapy + Radiotherapy + Surgery	9	12.9	2	2.8
Oral Mucositis	Present	22	31.4	-	-
	Absent	48	68.6	-	-

Source: Study data, 2017.

**Table 2 ijerph-16-03395-t002:** Results of the “performance” and “capacity” qualifiers for the mobility and self-care domains of the International Classification of Functioning, Disability, and Health (ICF). João Pessoa, PB, Brazil, 2017.

MOBILITY	Performance, *n* (%)					Capacity, *n* (%)				
	0	1	2	3	4	0	1	2	3	4
Altering the basic body position	62 (88.6)	7 (10.0)	0 (0.0)	1 (1.4)	0 (0.0)	41 (58.6)	14 (20.0)	12 (17.1)	2 (2.9)	1 (1.4)
Maintaining a body position	65 (92.9)	3 (4.3)	1 (1.4)	1 (1.4)	0 (0.0)	40 (57.1)	17 (24.3)	11 (15.8)	1 (1.4)	1 (1.4)
Self-transferring	67 (95.7)	2 (2.9)	0 (0.0)	0 (0.0)	1 (1.4)	55 (78.6)	7 (10.0)	6 (8.6)	1 (1.4)	1 (1.4)
Lifting and carrying objects	68 (97.2)	1 (1.4)	0 (0.0)	0 (0.0)	1 (1.4)	55 (78.6)	8 (11.4)	6 (8.6)	0 (0.0)	1 (1.4)
Moving objects with the lower limbs	70 (100.0)	0 (0.0)	0 (0.0)	0 (0.0)	0 (0.0)	69 (98.6)	1 (1.4)	0 (0.0)	0 (0.0)	0 (0.0)
Fine use of the hand	68 (97.1)	0 (0.0)	2 (2.9)	0 (0.0)	0 (0.0)	65 (92.8)	2 (2.9)	1 (1.4)	1 (1.4)	1 (1.4)
Use of the hand and arm	69 (98.6)	0 (0.0)	1 (1.4)	0 (0.0)	0 (0.0)	65 (92.8)	2 (2.9)	1 (1.4)	2 (2.9)	0 (0.0)
Walking	58 (82.9)	10 (14.3)	0 (0.0)	1 (1.4)	1 (1.4)	36 (51.5)	21 (30.0)	4 (5.7)	5 (7.1)	4 (5.7)
Moving around	56 (80.0)	8 (11.4)	3 (4.3)	2 (2.9)	1 (1.4)	30 (42.9)	20 (28.5)	9 (12.9)	5 (7.1)	6 (8.6)
Moving around different locations	60 (85.7)	6 (8.6)	2 (2.9)	1 (1.4)	1 (1.4)	46 (65.7)	10 (14.3)	6 (8.6)	5 (7.1)	3 (4.3)
Moving around using equipment	66 (94.3)	2 (2.9)	1 (1.4)	0 (0.0)	1 (1.4)	64 (91.4)	2 (2.9)	2 (2.9)	1 (1.4)	1 (1.4)
Using transportation (car, bus, train, plane etc.).	69 (98.6)	0 (0.0)	0 (0.0)	0 (0.0)	1 (1.4)	62 (88.5)	2 (2.9)	3 (4.3)	1 (1.4)	2 (2.9)
SELF-CARE										
Washing oneself (bathing, drying, washing hands etc).	68 (97.2)	1 (1.4)	0 (0.0)	0 (0.0)	1 (1.4)	43 (61.4)	10 (14.3)	14 (20.0)	2 (2.9)	1 (1.4)
Caring for body parts (brushing teeth, shaving etc.)	66 (94.3)	3 (4.3)	0 (0.0)	0 (0.0)	1 (1.4)	32 (45.7)	10 (14.3)	23 (32.9)	4 (5.7)	1 (1.4)
Dressing	68 (97.2)	1 (1.4)	0 (0.0)	0 (0.0)	1 (1.4)	40 (57.1)	17 (24.3)	12 (17.2)	0 (0.0)	1 (1.4)
Eating	67 (95.8)	1 (1.4)	0 (0.0)	1 (1.4)	1 (1.4)	63 (90.0)	3 (4.3)	1 (1.4)	1 (1.4)	2 (2.9)
Drinking	68 (97.2)	1 (1.4)	0 (0.0)	0 (0.0)	1 (1.4)	64 (91.4)	3 (4.3)	2 (2.9)	0 (0.0)	1 (1.4)
Caring for own health	62 (88.6)	5 (7.1)	2 (2.9)	0 (0.0)	1 (1.4)	11 (15.7)	18 (25.7)	16 (22.9)	18 (25.7)	7 (10.0)

Source: Study data, 2017.

**Table 3 ijerph-16-03395-t003:** Relationship between explanatory variables and their corresponding *p*-values in relation to the study outcomes (*p*-value < 0.3).

Variable	*p*-Value *
Age	0.894
Sex	0.585
Marital Status	0.575
Ethnicity/Race	0.178 *
Educational Level	0.038 *
School shift (morning, afternoon, double shift)	0.085 *
Monthly Family Income	0.527
Cancer Type	0.043 *
Treatment Modality	0.991
p410p Altering the Basic Body Position	0.138 *
p415p Maintaining a Body Position	0.216 *
p420p Self-transferring	0.992
p430p Lifting and Carrying Objects	0.994
p435p Moving Objects with the Lower Limbs	1.000
p440p Fine Use of the Hands	0.575
p445p Use of the Hand and Arm	0.991
p450d Walking	0.045 *
p455p Moving around	0.239 *
p460p Moving around Different Places	0.072 *
p465p Moving around Using Equipment	0.575
p470p Using Transportation (car, bus, train, plane etc.)	0.991
p475p Driving (bicycle, motorcycle, car etc.)	1.000
p480p Riding Animals for Transportation	1.000
p510p Washing Oneself (bathing, drying, washing hands etc.)	0.994
p520p Caring for Body Parts (brushing teeth, shaving etc.)	0.996
p530p Toileting	1.000
p540p Dressing	0.994
p550p Eating	0.994
p560p Drinking	0.994
p570p Caring for Own Health	0.573
p410c Altering the Basic Body Position	0.034 *
p415c Maintaining a Body Position	0.003 *
p420c Self-transferring	0.496
p430c Lifting and Carrying Objects	0.208 *
p435c Moving Objects with the Lower Limbs	0.991
p440c Fine Use of the Hands	0.575
p445c Use of the Hand and Arm	0.562
p450c Walking	0.041 *
p455c Moving around	0.041 *
p460c Moving around Different Places	0.175 *
p465c Moving around Using Equipment	0.480
p470c Using Transportation (car, bus, train, plane etc.	0.216 *
p475c Driving(bicycle, motorcycle, car etc.)	1.000
p480c Riding Animals for Transportation	1.000
p510c Washing oneself (bathing, drying, washing hands etc.)	0.575
p520c Caring for Body Parts (brushing the teeth, shaving etc.)	0.005 *
p530c Toileting	1.000
p540c Dressing	0.991
p550c Eating	0.996
p560c Drinking	0.216 *
p570c Caring for Own Health	0.044 *

* Univariate logistic regression. *p*-value < 0.3; c—Capacity; p—Performance. Source: Study data, 2017.

**Table 4 ijerph-16-03395-t004:** Significant variables in the final logistic regression model.

Variable	Parameter Estimation	Standard Error	*p*-Value *	OR	95% CI
Intercept	−1.0613	0.4554	0.019		
Age (07–10 years)	1.2866	0.7043	0.067	3.6204	[2.24; 5.0008]
Cancer Type (Solid tumor)	−1.8562	0.7351	0.011	0.1562	[−1.2845; 1.5969]
Maintain a Body Position (Capacity—severe difficulty)	2.6711	0.9467	0.004	14.4558	[12.6003; 16.3113]
Walking (Capacity—severe difficulty)	3.2358	1.3959	0.020	25.4267	[22.6908; 28.1626]

* Multiple Logistic Regression. Stepwise Backward Selection Method. Significant *p*-value if <0.05. OR: Odds Ratio, CI: Confidence Interval. Source: Study data, 2017.

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
