# Peer review of "The Contribution of Motor Changes to Oral Mucositis in Pediatric Cancer Patients: A Cross-Sectional Study"

_ijerph, 2019, doi:10.3390/ijerph16183395_

Round 1
Reviewer 1 Report
Lucena et al. used descriptive statistics and logistic regression methods to explore factors related to oral mucositis in pediatric patients with cancer. This study showed some interesting findings but the authors need to provide more details of the research design and the statistical methods. Especially, the format of this manuscript should be carefully revised and improved.
Major points:
1. Table 1. The variables were presented in a very confused way. E.g., why ‘No schooling
’ and ‘Elementary school’ belong to ‘Race/Ethnicity’? This table should be reformatted.
2. Table 2. What the meanings of ‘0’,’1’,’2’,’3’,’4’? How the classification was defined?
3. Also Table 2, why some showed ‘n’ and others showed ‘%’? This table should be reformatted too.
4. Table 3. ‘OR’ and ‘CI’ need to be spelled out.
5. Also Table 3, the authors need to describe how the p-values were calculated.
6. The authors need to revisit the variables carefully. For example, I don’t think the ‘Educational level’ truly related to oral mucositis. Only six participants were attending elementary school. This sample size is too small to make a reliable conclusion. Actually, the age of the patient and the course of cancer seem to be the ‘real’ factors related to this symptom, and the ‘Educational level’ happens to positively relate with age.
Minor points:
1. There are many missing spaces. E.g., ‘n=38’ should ‘n = 38’ (Line 107).
2. Line 107, ‘…with a mean age of 10.9 years (±4.90).’ Is this SD or SE?
Author Response
Reviewer #1
Lucena et al. used descriptive statistics and logistic regression methods to explore factors related to oral mucositis in pediatric patients with cancer.
This study showed some interesting findings but the authors need to provide more details of the research design and the statistical methods. Especially, the format of this manuscript should be carefully revised and improved.
Response: We appreciate the reviewer’s comments, which encouraged us to keep studying oral mucositis, a clinically relevant comorbidity afflicting children and adolescents with cancer. We have incorporated all suggestions into the manuscript and thoroughly revised the text as per request.
Table 1. The variables were presented in a very confused way. E.g., why ‘No schooling’ and ‘Elementary school’ belong to ‘Race/Ethnicity’? This table should be reformatted.
Response: We apologize for the confusion and inform that Table 1 has been corrected accordingly.
Table 2. What the meanings of ‘0’,’1’,’2’,’3’,’4’? How the classification was defined?
Response: We thank the reviewer for the scrutiny. These numbers correspond to the scores of the International Classification of Functioning, Disability and Health (ICF). An informative note has been added to Table 2.
Also Table 2, why some showed ‘n’ and others showed ‘%’? This table should be reformatted too.
Response: We inform that Table 2 has been revised and now shows absolute and relative frequencies as per request.
Table 3. ‘OR’ and ‘CI’ need to be spelled out.
Response: We inform that the terms OR and CI have been spelled out in Table 4 (former Table 3), as suggested.
Also Table 3, the authors need to describe how the p-values were calculated.
Response: Considering suggestions given by another review, the tables were reformatted, and former Table 3 stands now as Table 4. In Table 4, we included the information on how p-values were calculated, as suggested by the reviewer.
The authors need to revisit the variables carefully. For example, I don’t think the ‘Educational level’ truly related to oral mucositis. Only six participants were attending elementary school. This sample size is too small to make a reliable conclusion. Actually, the age of the patient and the course of cancer seem to be the ‘real’ factors related to this symptom, and the ‘Educational level’ happens to positively relate with age.
Response: Wetotally agree with the reviewer and inform that a new data analysis was carried out. We removed the variable “Educational Level” and introduced the variable “Age” along with the definition of new cutoff points based on previous reports of our research group.
(RIBEIRO, I.L.A.; LIMEIRA, R. R. T.; CASTRO, R. D.; BONAN, P. R. F.; VALENÇA, A. M. G.Oral mucositis in pediatric patients in treatment for acute lymphoblastic leukemia. INTERNATIONAL JOURNAL OF ENVIRONMENTAL RESEARCH AND PUBLIC HEALTH (PRINT), v. 14, p. 2-7, 2017.; DAMASCENA, L. C. L. ; LUCENA, N. N. N. ; RIBEIRO, I. L. A. ; PEREIRA, T. L. ; CASTRO, R. D. ; BONAN, P. R. F. ; LIMA NETO, E. A. ; LIMA FILHO, L. M. A. ; VALENÇA, A. M. G. . Factors Contributing to the Duration of Chemotherapy-Induced Severe Oral Mucositis in Oncopediatric Patients. International Journal of Environmental Research and Public Health, v. 15, p. 1153, 2018).
There are many missing spaces. E.g., ‘n=38’ should ‘n = 38’ (Line 107).
Response: We apologize for the missing spaces and inform that whole text has been revised accordingly.
Line 107, ‘…with a mean age of 10.9 years (±4.90).’ Is this SD or SE?
Response: This corresponds to the SD. This information has been included in the sentence.

Reviewer 2 Report
This cross-sectional study shows that presence of oral mucositis was associated with exercise performance among children/adolescents with cancer in a group of Brazilian.
This is an interesting intervention study. However, I would like to make some points regarding the manuscript. The article needs to be revised. First of all, the authors should follow the STROBE guideline totally and revise the whole text. Second, the authors should re-perform the logistic analyses. Third, there are no important data related with oral mucositis. Finally, the native speaker should check the grammar and typos (no space, strange line breaks, etc.).
TITLE
1) Please add the study design following the guideline (see above); i.e., “A cross-sectional study”.
ABSTRACT
1) The authors should change the abstract after re-analyzing data.
INTRODUCTION
1) The logic is unclear. Do the authors seek whether impairment of exercise performance may affect onset of oral mucositis or not? What is the mechanism? The oral mucositis may affect exercise performance (reversed). On the other hands, systemic conditions may both oral mucositis and exercise performance (outcome-outcome). Please add more comments and references in the introduction and discussion, respectively.
2) Please add the hypothesis before the aim.
MATERIALS AND METHODS
1) How did the author recruit the 70 patients? The flowchart should be shown.
2) Please add the study design and inclusion criteria at the early part.
3) Please add the comments about “written” or “verbal” consent in the text.
4) The number of case was only 22. Because of many valuables, the logistic model in this study seems to be inappropriate based on the previous report (J Clin Epidemiol Vol. 49, No. 12, pp. 1373-1379, 1996). Please re-analyze the data carefully (see above).
5) Did the authors check the status of self-care, professional oral care, oral conditions, postoperative complications, blood conditions, body weight, body temperature, nutrition, food style, renal function, and other important conditions? If no, it is very big limitation.
6) Please add the detail definition of mucositis in addition to the name of guideline, because it is very important in this study. How about the grade definition?
7) Please add the comments about how missing data were addressed.
8) Did the authors perform the sample size estimation?
RESULTS
1) The results should be changed by new analyses.
2) Can the authors add the flowchart according to the guideline? The authors should show the estimate number of a population (see above).
3) Please add a new Table 3 to only show the association between oral mucositis and other factors. Then, the current Tables 3 should be revised and move to a new Table 4. The authors should delete the current Table 4.
DISCUSSION
1) Please change the discussion after adding the new results.
2) The bruxism assessment did not fully match in this study with the consensus. Please delete the comments in the strength part.
3) Please add the comments about generalizability and/or specificity.
4) Please add the comments about limitation with appropriate references; i.e., a cross-sectional study, small number of patients, a single center study, weak methodology and no important data of confounders at least.
5) The conclusion should be shortened and changed after new analyses. Please deleted the sentences, “These findings indicate…well-being of these patients”, because this is just a cross-sectional study.
Author Response
Reviewer #2
This cross-sectional study shows that presence of oral mucositis was associated with exercise performance among children/adolescents with cancer in a group of Brazilian. This is an interesting intervention study. However, I would like to make some points regarding the manuscript. The article needs to be revised.
Response: We thank the reviewer for the scrutiny and inform that all modifications have been incorporated and the whole manuscript has been revised accordingly.
First of all, the authors should follow the STROBE guideline totally and revise the whole text.
Response: As per request, the STROBE guidelines were followed, and the text was properly revised: Title – now mentions the study design; Introduction – now includes the study hypothesis; and Methods – now includes the flowchart of sample selection.
Second, the authors should re-perform the logistic analyses.
Response: We thank the reviewer for the suggestion and inform that a new data analysis was
Third, there are no important data related with oral mucositis.
Response: We have included important information on oral mucositis in the Introduction and Discussion sections of the manuscript.
Finally, the native speaker should check the grammar and typos (no space, strange line breaks, etc.).
Response: We inform that the whole manuscript has been revised for English editing improvement.
TITLE
1) Please add the study design following the guideline (see above); i.e., “A cross-sectional study”.
Response: The title has been modified as per request.
ABSTRACT
The authors should change the abstract after re-analyzing data.
Response: The abstract has been modified based on the new data analysis.
INTRODUCTION
1) The logic is unclear. Do the authors seek whether impairment of exercise performance may affect onset of oral mucositis or not? What is the mechanism? The oral mucositis may affect exercise performance (reversed). On the other hands, systemic conditions may both oral mucositis and exercise performance (outcome-outcome). Please add more comments and references in the introduction and discussion, respectively.
Response: This is a very interesting point. We appreciate the reviewer’s comment and inform that additional references have been included and discussed in the manuscript. Please note that new hypotheses were raised for the associations observed in our study.
2) Please add the hypothesis before the aim.
Response: The hypothesis was placed before the aim of the study, as per request.
MATERIALS AND METHODS
How did the author recruit the 70 patients? The flowchart should be shown.
Response: Please note that a flowchart of study selection has been included to the manuscript (Figure 1).
Please add the study design and inclusion criteria at the early part.
Response: Information about the study design and inclusion criteria were included.
Please add the comments about “written” or “verbal” consent in the text.
Response: We inform that a sentence specifying that prior written consent was obtained was added to the text.
The number of case was only 22. Because of many valuables, the logistic model in this study seems to be inappropriate based on the previous report (J Clin Epidemiol Vol. 49, No. 12, pp. 1373-1379, 1996). Please re-analyze the data carefully (see above).
Response: We appreciate the reviewer’s scrutiny and the opportunity to reanalyze the data. We have included information showing the fitness and accuracy of the model obtained (deviance, ROC curve, true predictive values). We further note that the final model follows the recommendations of the article suggested by the reviewer (J Clin Epidemiol Vol. 49, No. 12, pp. 1373-1379, 1996), which indicates a minimum of 10 observations for each variable.
5) Did the authors check the status of self-care, professional oral care, oral conditions, postoperative complications, blood conditions, body weight, body temperature, nutrition, food style, renal function, and other important conditions? If no, it is very big limitation.
Response: This is a very interesting point. While we agree with the reviewer that some clinical parameters are relevant in the context of oral mucositis, particularly blood parameters (e.g., neutropenia) and renal function (e.g., elevated creatinine levels), we note that these variables are mostly related to more severe cases of oral mucositis. The primary purpose of our study was to determine the occurrence of oral mucositis (presence/absence) in cancer pediatric patients rather than its severity (mild, moderate, severe).
As for nutritional aspects, all study volunteers were on oral diet. We inform that this information has been included in the Discussion section.
The variables related to self-care were assessed according to the “Self-care” domain of the ICF. This domain encompasses the following items: Washing oneself (bathing, drying, washing hands etc); caring for body parts (brushing teeth, shaving etc); toileting; dressing; eating; drinking) and caring for own health. We note that the univariate analysis indicated an association between the occurrence of oral mucositis and “caring for body parts” (brushing teeth shaving, etc),“drinking” and “caring for own health”. Nevertheless, the association was no longer observed in the final model.
With regard to oral care, patients admitted to the pediatric ward of the hospital are assisted by a multidisciplinary research team. These professionals perform daily assessments of the patient’s oral conditions (based on OAG) and provide educational instructions as well as preventive and therapeutic approaches. Thus, all patients included in our study and their respective legal guardians (parents/caregivers) had access to educational activities (oral hygiene instruction and information on oral mucositis) and preventive measures (topical application of fluoride).
We further note that the first studies performed by our research group did not include clinical variables (other than the OAG), e.g. biofilm accumulation and gingival bleeding, as the examiners were not calibrated accordingly to these conditions, which may represent a limitation of our study.Nevertheless, in order to address the concern raised by the reviewer, we have included some limitations of the study at the end of the Discussion section.
Please add the detail definition of mucositis in addition to the name of guideline, because it is very important in this study. How about the grade definition?
Response: We have included the definition of oral mucositis in the Introduction section, and information on OAG and its grading system in the Material and Methods section.
Please add the comments about how missing data were addressed.
Response: We inform that there was no missing data as the data were collected directly from study participants. This information has been included in the text.
Did the authors perform the sample size estimation?
Response: We performeda posteriori sample size calculation. In the logistic regression model the Wald test statistic is given by the ratio between the maximum likelihood estimate of the parameter and the estimate of its standard error (SE). The test statistics, under the null hypothesis (H0: βj = 0), has standard normal distribution.
The p-value is defined by:
,
where Z follows a standard normal distribution. If the p-value was less than or equal to α (significance level adopted), the null hypothesis was rejected.
Sample size was determined by Whittemore approximation (Hsieh, 1989). According to the obtained prevalence and odds ratio values, for a sample size of 70, we obtained a significance level of 5% (α) and a power of 82.5% (1-β).
Hsieh, F. Y. Sample size tables for logistic regression, Statistics in Medicine, 1989, 8, pp. 795-802.
RESULTS
The results should be changed by new analyses.
Response: We inform that the Results section has been modified based on the new data analysis.
Can the authors add the flowchart according to the guideline? The authors should show the estimate number of a population (see above).
Response: We appreciate the reviewer’s recommendation and inform that a STROBE-based flowchart of study sample selection was included in the Material and Methods section.
Please add a new Table 3 to only show the association between oral mucositis and other factors. Then, the current Tables 3 should be revised and move to a new Table 4. The authors should delete the current Table 4.
Response: We totally agree with the reviewer’s suggestion. The original Table 4 was removed from the manuscript; the current Table 3 was revised to include all explanatory variables and p-values of the univariate analysis. Table 3 was renumbered and now stands as Table 4.
DISCUSSION
Please change the discussion after adding the new results.
Response: The Discussion section was rewritten based on the new data analysis.
The bruxism assessment did not fully match in this study with the consensus. Please delete the comments in the strength part.
Response: We believe there might have been some misunderstanding, as bruxism was not addressed in our study.
Please add the comments about generalizability and/or specificity.
Response: This is a relevant point. We included additional comments stating when caution is needed while interpreting the results. The study was carried out in a single hospital (not a multicenter study), and further research is needed to consider other variables related to the topic.
Please add the comments about limitation with appropriate references; i.e., a cross-sectional study, small number of patients, a single center study, weak methodology and no important data of confounders at least.
Response: We appreciate the suggestion and inform that study limitations has been included in the manuscript along with their corresponding citations.
The conclusion should be shortened and changed after new analyses. Please deleted the sentences, “These findings indicate…well-being of these patients”, because this is just a cross-sectional study.
Response: We inform that the conclusion has been rephrased based on the reviewer’s suggestions.

Round 2
Reviewer 1 Report
The authors addressed all my questions very well. I suggest to accept this manuscript.
Author Response
Reviewer #1
Comments and Suggestions for Authors
The authors addressed all my questions very well. I suggest to accept this manuscript.
Response: We appreciate the reviewer’s comments and all suggestions given to improve the quality of our manuscript.
Reviewer 2 Report
The paper was overall improved. However, there are some issues. The paper should be revised.
ABSTRACT
1) Please insert a space between “with” and “70” (L15).
2) Please add the words, “Within the limitations of this study, “ in the conclusion because the paper has big limitations.
INTRODUCTION
1) Please insert a space between ” hormones” and “(“ (L61). Please also check the other typos very carefully.
MATERIALS AND METHODS
1) Did the authors mean that “a sample size of 70” (L127) is a number in each group or 35 vs. 35?
2) The authors should use a stepwise backward selection logistic regression model.
RESULTS
1) In the Figure 1, why did the authors exclude Individuals diagnosed with cancer (n = 26) among the targeted patients who were diagnosed with any type of cancer?
2) Please add the number of participants in each category [eg. Age (07-10 years); presence of oral mucositis, n=? (?%)] in the Table 3. There are only p-values. Furthermore, please add the name of statistical analysis in the footnote.
3) There were no references in the Table 4. Please add the reference in each category. Furthermore, the authors should revise the results using a stepwise backward selection logistic regression model (see above).
DISCUSSION
1) Please add the words, “Within the limitations of this study, “ in the conclusion.
2) Please delete “In addition to oral care, maintaining functionality in children and adolescents with cancer is essential to reduce the prevalence and severity of oral mucositis, improve their oral health status, and avoid interruptions in the antineoplastic treatment”.
Author Response
Reviewer #2
Comments and Suggestions for Authors
The paper was overall improved. However, there are some issues. The paper should be revised.
Response: We thank the reviewer for the additional suggestions and inform that these have been incorporated into the manuscript.
ABSTRACT
1) Please insert a space between “with” and “70” (L15).
Response: We inform that the whole manuscript has been revised for issues like this.
2) Please add the words, “Within the limitations of this study, “ in the conclusion because the paper has big limitations.
Response: Please note that the Abstract and Conclusion now contain the sentence suggested by the reviewer.
INTRODUCTION
1) Please insert a space between ” hormones” and “(“ (L61). Please also check the other typos very carefully.
Response: We thank the reviewer for the scrutiny and inform again that the whole manuscript has been revised for these typos.
MATERIALS AND METHODS
1) Did the authors mean that “a sample size of 70” (L127) is a number in each group or 35 vs. 35?
Response: We meant a single group consisting of 70 patients. This information has been made clear in the text.
2) The authors should use a stepwise backward selection logistic regression model.
Response: We did. Please note that this information was more clearly described in the end of the Methods section and in the footnote of Table 4.
RESULTS
1) In the Figure 1, why did the authors exclude Individuals diagnosed with cancer (n = 26) among the targeted patients who were diagnosed with any type of cancer?
Response: We appreciate the reviewer’s comment.There was a typo as we actually meant “patients whose diagnosis was negative for cancer”. Please note that this information was corrected in Figure 1.
2) Please add the number of participants in each category [eg. Age (07-10 years); presence of oral mucositis, n=? (?%)] in the Table 3. There are only p-values. Furthermore, please add the name of statistical analysis in the footnote.
Response: The reviewer gave a very relevant suggestion to improve our data presentation.Please see that Table 1 now shows all the absolute and relative frequencies of the variables in relation to the presence of oral mucositis. Moreover, the name of the statistical test was included in the footnote.
3) There were no references in the Table 4. Please add the reference in each category. Furthermore, the authors should revise the results using a stepwise backward selection logistic regression model (see above).
Response: References were included in each category as suggested by the reviewer. In addition, we further described that the stepwise backward selection method was used for data analysis.
DISCUSSION
1) Please add the words, “Within the limitations of this study, “ in the conclusion.
Response: Please note that the Abstract and Conclusion now contain the sentence suggested by the reviewer.
2) Please delete “In addition to oral care, maintaining functionality in children and adolescents with cancer is essential to reduce the prevalence and severity of oral mucositis, improve their oral health status, and avoid interruptions in the antineoplastic treatment”.
Response: This sentence has been removed from the manuscript as per request.